# Monkeys can identify pictures from words

**Elizabeth Cabrera-Ruiz**[1,2☯], **Marlen Alva**[1☯], **Mario Treviño**[3], **Miguel Mata-Herrera**[1], **José Vergara**[4], **Tonatiuh Figueroa**[1], **Javier Perez-Orive**[2], **Luis Lemus**[1]*

**1** Department of Cognitive Neuroscience, Instituto de Fisiología Celular, Universidad Nacional Autónoma de México, Mexico City, México, **2** Basic Neurosciences, Instituto Nacional de Rehabilitacion, "Luis Guillermo Ibarra Ibarra" Mexico City, México, **3** Laboratorio de Plasticidad Cortical y Aprendizaje Perceptual, Instituto de Neurociencias, Universidad de Guadalajara, Guadalajara, Jalisco, México, **4** Department of Neuroscience, Baylor College of Medicine, Houston, TX, United States of America

☯ These authors contributed equally to this work.
* lemus@ifc.unam.mx

**Data Availability Statement:** All relevant data for this study is fully available without restrictions from the Figshare repository. The dataset can be accessed at the following DOI: https://doi.org/10.6084/m9.figshare.27111460.

## Abstract

Humans learn and incorporate cross-modal associations between auditory and visual objects (e.g., between a spoken word and a picture) into language. However, whether non-human primates can learn cross-modal associations between words and pictures remains uncertain. We trained two rhesus macaques in a delayed cross-modal match-to-sample task to determine whether they could learn associations between sounds and pictures of different types. In each trial, the monkeys listened to a brief sound (e.g., a monkey vocalization or a human word), and retained information about the sound to match it with one of 2–4 pictures presented on a touchscreen after a 3-second delay. We found that the monkeys learned and performed proficiently in over a dozen associations. In addition, to test their ability to generalize, we exposed them to sounds uttered by different individuals. We found that their hit rate remained high but more variable, suggesting that they perceived the new sounds as equivalent, though not identical. We conclude that rhesus monkeys can learn cross-modal associations between objects of different types, retain information in working memory, and generalize the learned associations to new objects. These findings position rhesus monkeys as an ideal model for future research on the brain pathways of cross-modal associations between auditory and visual objects.

## Introduction

Humans form cross-modal associations (CMAs) between sounds and images, which play a vital role in integrating semantic representations within language [1]. Supporting this, fMRI studies have shown that the temporal lobe of the human brain is actively involved in CMAs [2, 3] between words and visual objects [4]. It is believed that CMAs between phonological "templates"—developed in human infants by listening to caretakers—and observed objects are essential for creating semantic representations and aiding the production of a child's first words [5–8]. Similarly, auditory templates have been proposed as a mechanism for vocal production in birds [9–13] and marmoset monkeys [14]. Recent studies, such as those by Carouso-Peck and Goldstein [15, 16], have also shown that visual signals during social

**Funding:** LL received funding from the Consejo Nacional de Humanidades, Ciencias y Tecnologías (CONAHCYT; Grant Number: 256767; https://conahcyt.mx/) and the Programa de Apoyo a Proyectos de Investigación e Innovación Tecnológica (PAPIIT; Grant Number: IN229323; https://dgapa.unam.mx/index.php/impulso-a-la-investigacion/papiit). JV was supported by the Secretaría de Educación, Ciencia, Tecnología e Innovación de la Ciudad de México (Grant Number: SECTEI/103/2022; https://www.sectei.cdmx.gob.mx/). Elizabeth Cabrera Ruiz conducted this study to fulfill the requirements of the Programa de Doctorado en Ciencias Biomédicas at Universidad Nacional Autónoma de México and received a doctoral scholarship from the Consejo Nacional de Humanidades, Ciencias y Tecnologías (Scholarship Number: 245771; https://conahcyt.mx/). The data presented in this work form part of her doctoral dissertation. The funders had no role in study design, data collection and analysis, decision to publish, or preparation of the manuscript.

**Competing interests:** NO authors have competing interests.

interactions can also influence vocal production in birds. However, only a few ethological studies have suggested the existence of CMAs between vocal sounds and visual cues for semantic communication [17]. For instance, research has observed that vervet monkeys respond to calls signaling the presence of predators by looking upwards, downwards, or climbing into trees [18].

Neurophysiological recordings in monkeys have shown that the prefrontal cortex (PFC)—a brain area homologous to that in humans—utilizes working memory (WM) circuits [19] to perform CMAs between voices and faces [20–32], receiving inputs from various sensory regions [33–36]. CMAs have also been observed in the auditory and visual areas of the temporal lobe [37–45]. Notably, trained macaques have demonstrated the ability to perform cross-modal discriminations between visual and tactile objects [46, 47], and between stimuli that could be considered non-ethologically relevant (NER), such as between pitch and color [48] and between amodal information (i.e., information that does not belong to a particular modality) [49] such as numerosity [50] and flutter frequencies [51–54]. However, it remains to be explored whether non-human primates can establish CMAs between NER stimuli that are important for human language, like words—which monkeys can discriminate phonetically [55, 56], and pictures.

Therefore, to assess whether monkeys can form CMAs between NER stimuli, we trained two rhesus macaques in a delayed crossmodal match-to-sample task (DCMMS). We specifically designed the task to temporally separate auditory and visual stimuli, thus engaging WM circuits to retain one modality in mind while awaiting the corresponding cross-modal stimulus. Unlike prior studies, this task required the monkeys to retain auditory information during a 3-second WM period and then use this information to select a matching visual from a set of 2–4 pictures displayed simultaneously on a screen after the delay.

Our results show that rhesus monkeys can accurately identify sounds produced by various emitters and match them with images despite the temporal gap, highlighting the crucial role of WM circuits not only for storing information but also for actively evaluating the equivalence between stimuli of different sensory modalities. This finding suggests substantial similarities with human cognitive processing in analogous tasks [57, 58] and paves the way for future neurophysiological studies focused on identifying the specific brain pathways and mechanisms involved in these cross-modal processes.

## Materials and methods

### Ethics statement

Animal welfare was a priority throughout the study, conducted in strict accordance with the recommendations of the Official Mexican Norm for the Care and Use of Laboratory Animals (NOM-062-ZOO-1999). The protocol was approved by UNAM's IACUC (i.e., Comité Institucional para el Cuidado y Uso de Animales de Laboratorio; CICUAL; Protocol number: LLS200-22). Descriptions comply with the ARRIVE recommended guidelines [59]. The portrayal of one of the authors of this manuscript was used in the experiments and has given written informed consent to publish this case details.

### Subjects

Two adult rhesus monkeys (*Macaca mulatta*), a 10-year-old female (monkey G, 7 kg) and a 12-year-old male (monkey M, 12 kg) participated in the experiments. The animals had no previous training in any other task and were not subjected to any surgery or head restraint for this behavioral study. We adhered to the 3R principles (Replacement, Reduction, Refinement) [60]; therefore, we achieved statistical significance for the study in the number of trials each

monkey performed rather than in the number of animals employed. The monkeys were housed in cages in a temperature-controlled room (22˚C) with filtered air and day/night light cycles. They had free access to a balanced diet of dry food (pellets) supplemented with nuts, fresh fruits, and vegetables. Regular weight monitoring and veterinary check-ups ensured their health and well-being. The monkeys also had access to an enriched environment with toys, a recreation area for climbing and socializing with other monkeys four days a week, and opportunities for grooming through mesh sliding doors. In addition, cartoons and wildlife videos of content unrelated to the experiments, were presented on TV for no more than four hours a day. However, the face and voice of one of the researchers with whom the monkeys interacted were used during the experiments. To motivate participation in the experiments, the monkeys followed a water restriction protocol for 12–15 hours before experimental sessions (Monday to Friday, with water intake of 20–30 ml/kg achieved during the experimental sessions and ad libitum on weekends). After the 2–3-hour experimental sessions, they received 150g rations of fruits and vegetables.

## Experimental setup

The monkeys were trained to leave their cages and sit in a primate chair (Crist Instrument, INC.) for transfer to a soundproof booth adjacent to the vivarium for the experiments. The chair faced a touchscreen (ELO 2201L LED Display E107766, HD wide-aspect ratio 22in LCD) positioned 30 cm in front. A spring lever below the touchscreen (ENV-610M, Med Associates) allowed the monkeys to initiate the trials. Two speakers were mounted above the touchscreen: a Yamaha MSP5 Studio (40 W, 0.050–40 kHz) and a Logitech speaker (12 W, 0.01–20 kHz). These speakers delivered the sounds and background noise at 45- and 55 dB SPL, respectively. The monkeys received liquid rewards through a stainless-steel mouthpiece attached to the chair (Reward delivery system 5-RLD-E2-C Gravity feed dispenser, Crist Instrument INC.).

## Acoustic stimuli

The experiment utilized a variety of sounds, including laboratory recordings of words and monkey vocalizations, as well as free online sounds of cow vocalizations (https://freesound. org/). The sounds were edited to a duration of 500ms, resampled to 44.1 kHz (with cutoff frequencies of 0.1–20 kHz), and finally normalized (RMS) with Adobe Audition® 6.0 software. The phonetic labels of words in Spanish in the text and figure legends were created using the Automatic Phonetic Transcription tool by Xavier López Morras (http://aucel.com/pln/ transbase.html).

## Visual stimuli

The visual stimuli consisted of a red oval, grayscale cartoons of cows and monkeys, and pictures of human and monkey faces, as well as a cow circumscribed in ovals with a resolution of 200px/sq inch. Animal pictures used in the experiment were downloaded from free online sites and customized. However, the pictures shown in figures and supplementary information are similar but not identical to the original images used in the study; they were created for illustrative purposes only using an online AI image generator (https://www.fotor.com/ai-art-generator).

## Delayed crossmodal match-to-sample task

We trained two rhesus macaques in a DCMMS task to assess their ability to establish CMAs between sounds and images temporally decoupled. Each trial began with a 1˚ white cross

appearing in the center of the touchscreen. In response to the cross, the monkeys had to press and hold down a lever so that a 0.5-second reference sound could be delivered. After hearing the sound, the animals had to wait during a 3-second delay until 2–4 four pictures were presented simultaneously at random positions but equidistant on an 4˚ radius from the center of the touchscreen. The monkeys were then allowed to release the lever and select, within a 3-second response window, the picture that matched the sound (**S1 Video**). Correct selections were rewarded with a drop of liquid.

After the monkeys learned the task (see the monkeys' training section below), they were able to perform at different CMAs. Each CMA was established by associating a sound with a picture representing the same category of external stimulus (e.g., both corresponding to a human). For example, a CMA of the type 'human' consisted of the association between the word [si] and a human face. This way, CMAs of different types were created (e.g., monkey, cow, human, and color). In some cases, the monkeys associated a single sound with several pictures of the same type; for example, four monkey faces were associated with one 'coo', resulting in four 'monkey' CMAs (**S1 Table**). Each CMA at **S1 Table** was established by the monkeys after many sessions of practice (see the following methods' sections). However, in an experimental condition which we designated as the 'perceptual invariance experiment', we explored the monkeys' ability to recognize sounds uttered by different individuals the monkeys did not hear before the experiment. For example, a 'monkey' CMA substitution set was comprised of ten different coos (i.e., auditory versions uttered by different individuals) delivered randomly in different trials, but all those trials presented the same monkey picture as a match. Finally, all experimental sessions consisted of blocks of ~300 trials of intermixed CMAs. The Hit rate (HR) corresponds to the proportion of correct responses (i.e., audio-visual match) in a session; false alarms (FA) indicate the proportion of incorrect responses. Reaction times (RT) are the times to release the lever in response to the appearance of the pictures on the touchscreen. Motor times are the intervals between the lever release and the touching of the screen. The task was programmed using LabVIEW 2014 (64-bit SP1, National Instruments®). The artwork in the task description was created using a free online platform (https://www.fotor.com/ai-art generator).

## Monkeys training

To enhance the monkeys' speed and efficiency in learning the DCMMS task, we tailored stimuli, durations, and rewards according to their ongoing performance. Initially, the animals were trained to produce the motor responses necessary for the task, such as pressing and releasing a lever and consistently activating the touchscreen. Rewards were given for holding down the lever when the cross appeared at the center of the touchscreen and for releasing the lever and touching the screen upon its disappearance. After the subjects completed more than 90% of the trials in consecutive sessions, we introduced a gray filled circle on the touchscreen that appeared at random positions, requiring the monkey to touch it to receive a reward. Within one or two weeks, the animals consistently reacted to the cross within a 500 ms window of appearance, maintained the lever pressed for 5–7 seconds, and released it upon the disappearance of the cross to touch the visual target.

In the subsequent training phase, the monkeys were required to respond to a tone (i.e., a 0.5-second 440Hz; 55 dB SPL) randomly emitted from speakers on either side of the screen. The goal was to indicate the direction of the sound by touching a right or left circle on the screen, which appeared simultaneously with the tone and then after a gradually increasing delay (from 1–3 seconds). Here, the objective was for the monkeys to associate the auditory and visual locations. However, after more than 35,000 trials (i.e., ~ 117 sessions), the

performance remained at chance level. Consequently, we adopted a new approach that involved helping the monkeys to directly associate audio cues with specific images.

We replaced one circle with a cartoon image of a cow and added a 0.5-second broadband noise, so each trial featured either the tone or the noise. Rewards were given for correctly associating the cow cartoon with the broadband noise and the gray circle with the 440 Hz tone. From then on, sounds were delivered exclusively from a central speaker above the screen, and pictures appeared at different positions but were consistently separated by 180˚ (of visual angle) from each other. With this new training method, it took only a few sessions for Monkey G to begin performing above chance in associating the broadband noise with the cow cartoon (S1 Fig, upper leftmost panel). With many practice sessions, performance improved above chance, prompting us to gradually introduce new sounds and images to establish various CMAs. The initial CMAs involved only two different pictures on the touchscreen, while more complex associations involved the simultaneous presentation of three or four pictures.

## Learning measurements

Although the primary goal of our experiments was not to explore the learning process of macaques, we noted behavioral improvements throughout the sessions we aimed to document. To quantify this, we fitted learning curves to the performance at each CMA across sessions, thereby assessing the monkeys' learning progress. For this analysis, we applied the Rescorla-Wagner model, a well-established framework in associative learning [61], which explains learning as the formation of associations between conditioned and unconditioned stimuli. The process of deriving the learning curves required solving the following ordinary differential equation:

$$\frac{dV}{dt} = \alpha\beta(\lambda - V) \qquad \text{Eq(1)}$$

This equation describes the progression of associative strength ($V$) in response to trained conditioned stimuli, dependent on the number of training trials ($t$). The model provided the parameters for this equation: $\alpha\beta$, which is the product of the salience of the conditioned stimuli and the strength of the unconditioned stimuli (assumed constant during training, though modifications are possible [62]), and $\lambda$, representing the maximum possible associative strength towards the unconditioned stimulus. From the learning curves derived from this model, we extracted three additional parameters. Y0 measured initial performance, representing the starting point of the curve along the Y-axis. Parameter $\gamma$, indicating statistical learning onset, was determined as the first session in which performance reliably exceeded chance, defined as surpassing two standard deviations from the mean probability of a correct response under a binomial distribution (where p = chance level, and n = average number of trials per session). Finally, the derivatives of these learning curves, coupled with predefined thresholds, allowed us to determine the 'trend-to-optimal' experimental session for each CMA ($\delta$), marking the session where changes in performance from one session to the next did not exceed a designated minimal rate of improvement of y' = 0.01, indicating an approach towards a learning plateau.

## Statistical analysis

We focused most of our analyses on data collected post-training, after the monkeys' performance reached an asymptotic level, with their choices consistently exceeding the chance level. We used various statistical tests to compare RTs across different conditions. These tests included Spearman rank correlations to test the relationship between reaction time

distributions and the number of pictures on the touchscreen and a Kruskal-Wallis test for differences between CMAs. If the Kruskal-Wallis's test indicated a significant difference, we followed up with Mann-Whitney tests to compare conditions such as trials having 2 or 3 pictures when the same sound was presented. Finally, Bonferroni *post hoc* tests were used for multiple comparisons. The monkeys' chance performance threshold depended on the number of pictures displayed; for monkey M, the chance was 0.5 since it was performed in two picture sets only, while for monkey G it was 0.25 at four picture sets. Analyses were performed using MATLAB R2022 (MathWorks).

## Results

To investigate the ability of two rhesus monkeys to form CMAs between auditory and visual stimuli, we engaged them in a DCMMS task. Each trial commenced with the monkeys hearing a reference sound, followed by a 3-second delay, after which 2–4 pictures were displayed on the screen. Their task was to identify on the touchscreen the picture that corresponded to the sound (**Fig 1A**). The monkeys mastered fourteen CMAs after associating six distinct sounds—including broadband noise, animal vocalizations like a coo and a moo, and words such as ['tsan. gi], [si], and ['ro. xo]—with fourteen images (**S1 Table**). The trials varied, presenting either 2–4 pictures for Monkey G, while consistently presenting 2 pictures for Monkey M. Illustrative examples of four CMAs are depicted in **Fig 1B**. **Fig 1C and 1D** shows the monkeys' hit rate (HR) and false alarm rate (FA) across these CMAs (**S2 Table**). For example, when a coo sound was used as the reference, Monkey G correctly matched it with the monkey face 87.43% of the time, while its most frequent incorrect choice was the cow face, selected 5.94% of the time (**Fig 1C**, open boxplots). Overall, Monkey G exhibited a HR of 85.12% ± 9.11 (mean ± SD), and Monkey M achieved a HR of 87.07% ± 5.71. Statistical analysis showed no bias in their selection of specific positions on the touchscreen (one-way ANOVA with multiple pairwise comparisons; Tukey's HSD, $p < 0.05$) (**S2 Fig**). These outcomes indicate that both monkeys proficiently learned to discriminate each sound, against 2 to 4 pictures.

### Rhesus monkeys can learn cross-modal associations between stimuli of different types

The monkeys successfully established each CMA after several sessions of engaging in the DCMMS task, during which we initially presented two pictures simultaneously; only one of which corresponded to the played sound. To investigate the learning dynamics, we measured four learning parameters derived from fitting simple associative learning curves to the performance data across sessions. These parameters included the HR in the first session (y0), and the sessions marking statistical learning ($\gamma$), increasing learning ($\delta$), and asymptote of learning ($\lambda$), respectively (refer to Methods for detailed descriptions). The left panel in **Fig 2A** illustrates Monkey G's performance for the CMA between the coo sound and the monkey cartoon across sessions. Initially, the performance before learning was at chance level (~300 trials; see methods section on monkeys' training and learning measurements), aligning with the intersection of the learning curve (black line) and the Y-axis, termed Y0. Subsequently, the $\gamma$ performance level was reached after eight sessions from Y0 (~2700 trials); this level is defined as the session when the HR was above chance, marked by the intersection between the left edge of the gray box and the learning curve. A consistent increase in HR continued until the 15th session, reaching $\delta$ performance (right edge of the gray box), and by approximately the 40th session, the performance stabilized at the $\lambda$ level, where changes in performance from one session to the next were insignificant. Similarly, middle and right panels in **Fig 2A** show two CMAs learned in trials ending with 3 and 4 pictures, respectively.

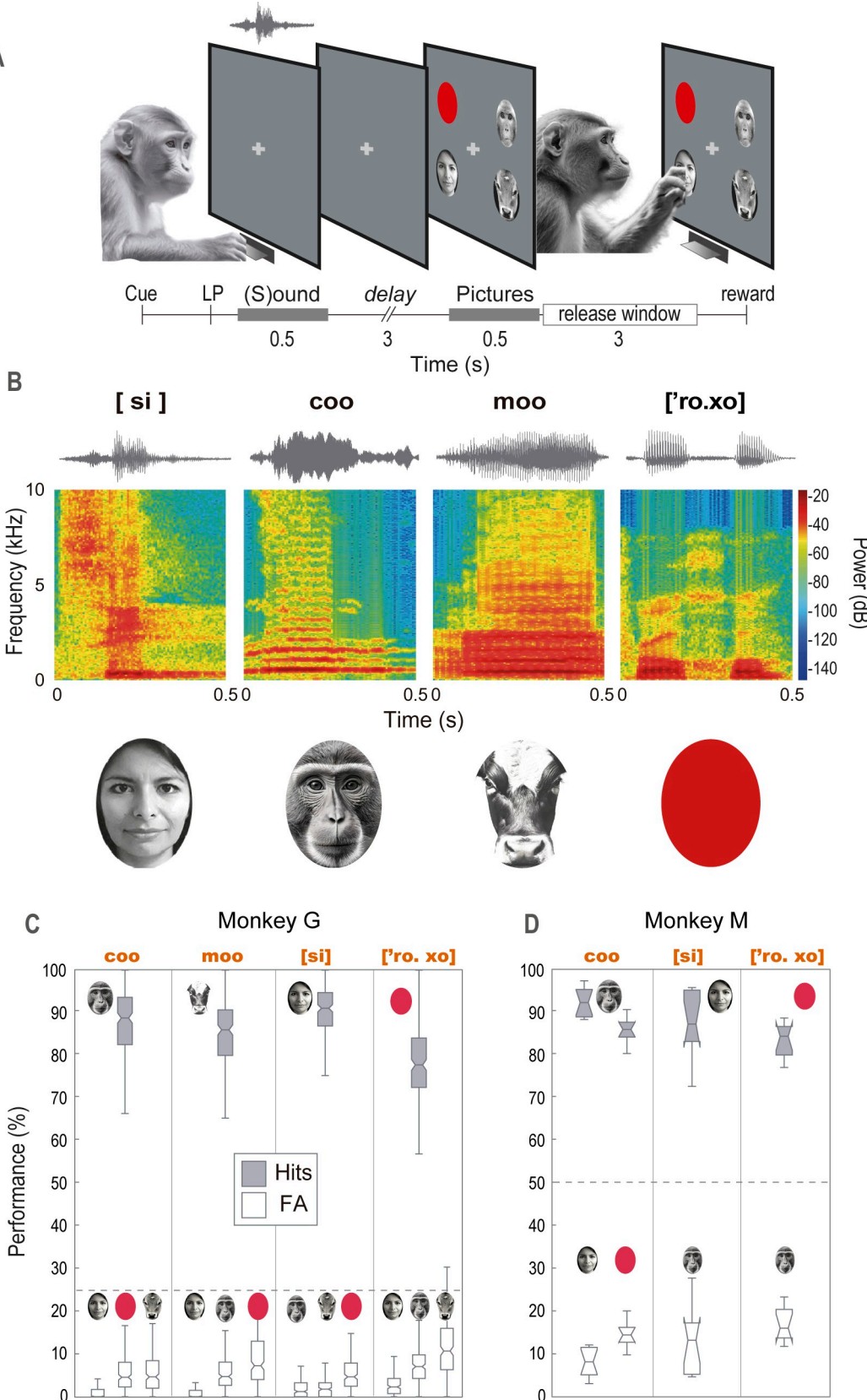

Fig 1. **Delayed crossmodal match-to-sample task.** (**A**) Task Events. A trial begins with the monkey pressing a lever in response to a cross appearing in the center of the touchscreen. This is followed by a 0.5-second reference sound, succeeded by a 3-second delay. After the delay, 2–4 pictures are simultaneously presented on the touchscreen. The monkey must then release the lever and touch the picture that matches the sample sound to receive a reward. LP indicates lever press. (**B**) Examples of Crossmodal Associations. Each column displays a CMA between a sound, represented visually by its sonogram and spectrogram and a picture. The sounds, marked in black, include two Spanish words (in IPA notation) and vocalizations of a monkey and a cow. (**C**) HR (close boxplots) and FAs (open boxplots) during the presentations of the CMAs shown in **B**. The dashed line indicates the performance at chance level (i.e., 25% for sounds discriminated against four pictures). The reference sound is labeled in red at the top of the graph. (**D**) Same as in **C**, but for Monkey M. The dashed line is set at the 50% chance level (i.e., two pictures on the screen). The pictures are similar but not identical to the original images used in the study and are therefore for illustrative purposes only.

S1 Fig shows performance evolving at each CMA across sessions in monkey G. In addition, the number of sessions needed for reaching sustained performance (i.e., the δ parameter) decreased in most new CMAs as the monkeys learned the aim of the task (Fig 2B). However, for the 'color' CMA formed by the word [ro. xo] (Spanish for 'red') and the red oval, Monkey

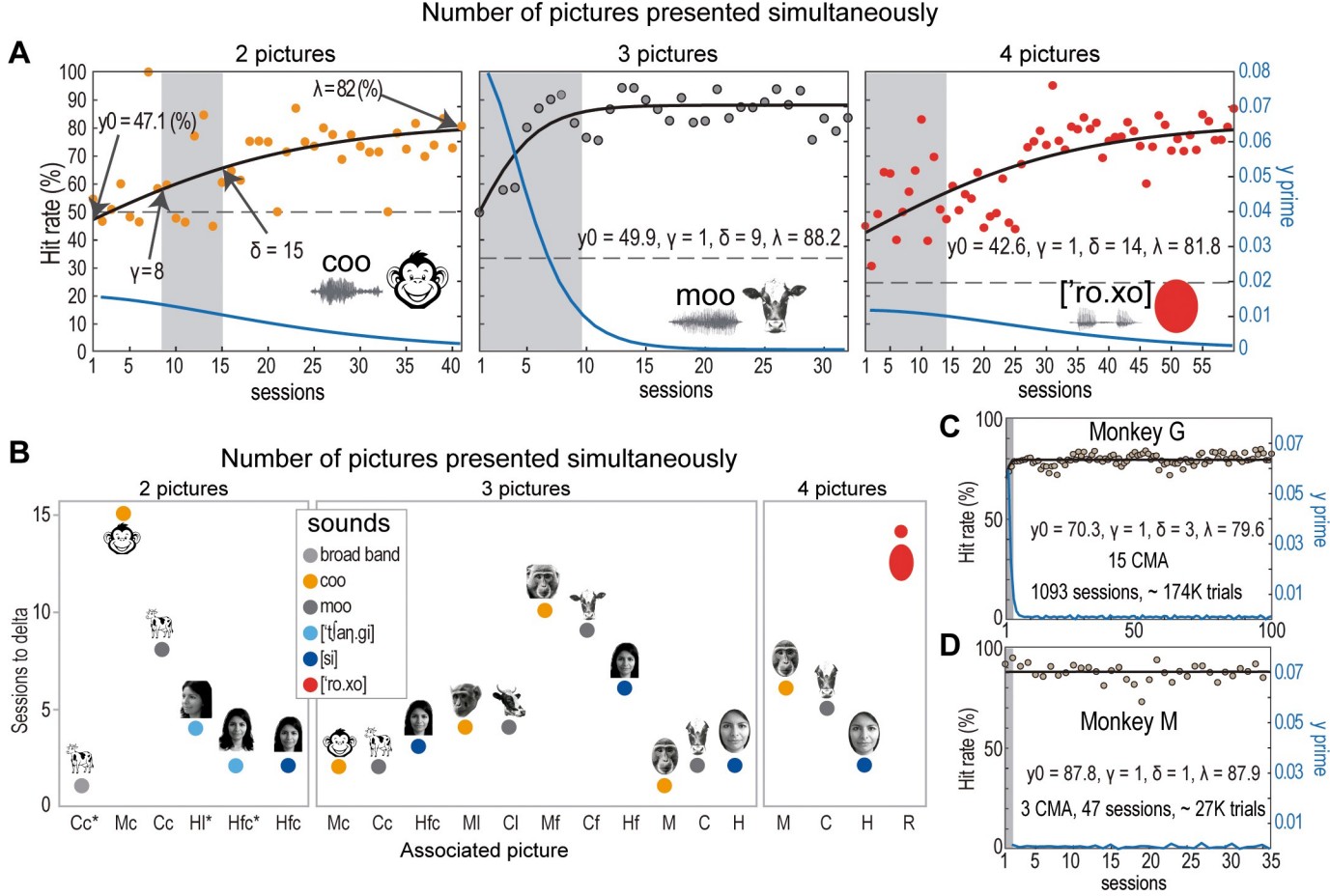

Fig 2. **Learning CMAs in monkeys.** (**A**) Monkey G's learning progress for three CMAs across sessions with trials presenting 2, 3, or 4 pictures simultaneously on the screen. The black line represents the average performance across sessions, while the blue line maps the first derivative of performance over training sessions (y' values), illustrating the rate of change at each session. The initial HR (Y0) was near chance level (indicated by the black line at the ordinates), followed by γ (the left edge of the gray box), where the HR statistically exceeded chance. The learning parameter δ, marks a period when HR increased consistently above chance, culminating in a performance plateau at the session denoted by the asymptote of learning λ. (**B**) Sessions before δ for each CMAs. (**C**) Represents the average performance of Monkey G across all CMAs over the sessions. (**D**) Same as in **C**, but for Monkey M. The pictures are similar but not identical to the original images used in the study and are therefore for illustrative purposes only.

G spent ~14 sessions to reach δ at the conditions where four pictures appeared on the screen. We interpret this increase in learning sessions as the result of introducing those stimuli for the first time in trials that presented four pictures on the screen. Finally, **Fig 2C and 2D** present the mean HR for all CMAs across sessions for both monkeys. We interpret the reduction in γ and δ as the monkeys solving the cognitive control of the motor behavior required for the task (procedure memory), e.g., pressing and releasing the lever and interacting correctly with the touchscreen, so that once this was done, the animals could focus only on learning the CMA associations.

## The RTs increased as a function of the selected picture and number of pictures on the touchscreen

To explore how different sounds and pictures influenced the monkeys' ability to find a cross-modal match, we analyzed the RTs during hits across various CMAs. **Fig 3A** displays Monkey G's RTs and motor times (MT) distributions across four CMAs. Notable differences are observed between the RT distributions, which pertain to the decision-making period (i.e., the time taken to decide which picture on the touchscreen matches the sound before releasing the lever). In contrast, the MT distributions, which relate to the stereotyped arm movement toward the chosen picture, showed no differences.

To assess whether acoustic or visual information primarily influenced the monkeys' RT distributions, we analyzed RTs to different pictures associated with a single sound (**S3 Table**), in trials presenting 3 pictures simultaneously. For instance, **Fig 3B** shows Monkey G's RT distributions (right panel) during correct responses to various pictures of the type 'monkey' (left panel) associated with a single 'coo'. **Fig 3C** shows the same for pictures of the type 'cow' associated with a single 'moo' sound. The RT distributions differed significantly in both instances ($p < 0.001$, Kruskal-Wallis's test), indicating that since the sounds were constant, the differences in RTs must have stemmed from variations among the pictures. This trend continued across all CMAs where different pictures were associated with the same sound ($p < 0.001$ for all comparisons, post hoc Mann-Whitney U tests with Bonferroni correction); pairwise comparisons between all pictures with each sound revealed significant differences in RT distributions ($p < 0.01$ for 71.43% of coo comparisons, 76.19% for moo comparisons, and 82.14% for [si] comparisons; Mann-Whitney U tests with Bonferroni correction). A similar effect is observed for FAs as shown in **Fig 3B** (insets), where the differences in RTs resulted from incorrect matchings ($p < 0.001$, Kruskal-Wallis's test).

Furthermore, **Fig 3D** shows that both the mean and the standard deviation (STD) of the RT distributions increased with the number of pictures displayed on the screen (2–4 pictures), indicating that locating the crossmodal match took longer as the number of distractor pictures increased. This tendency aligns with Weber's Law and studies in time processing [63]. Here, we interpret that the variation in STDs suggests that the faster RTs likely occurred when the matching picture was found first among the pool of pictures on the screen, and longer RTs when the match was found last. Notably, these variations in RTs did not impact the accuracy across different CMAs (**Fig 3E**). These findings imply that RT was more heavily influenced by the amount of visual information processed than by differences in sounds.

## The monkeys recognized sounds uttered by different speakers

We explored whether the monkeys could recognize sounds of the same type they learned but uttered by different individuals they did not hear before (**Fig 4A**, **S4 Table**). **Fig 4B** show how Monkey G performed above the 25% chance level in 98.33% of cases (paired-sample t-test, $p < 0.05$). Notably, the RTs during correct responses grouped into the four CMAs (i.e.,

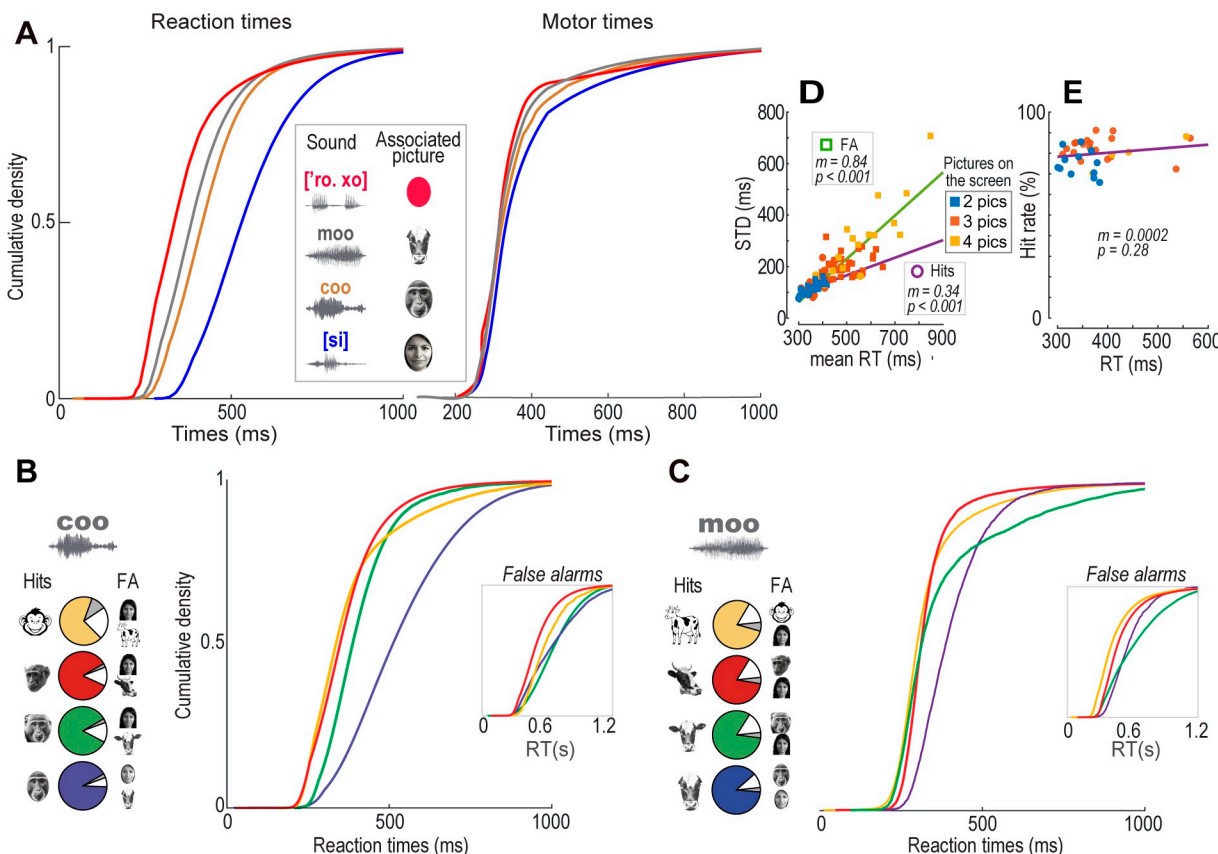

**Fig 3. Crossmodal associations influenced the monkeys' reaction times.** (A) Cumulative probabilities of reaction and motor times across four CMAs. (B) **Left panel,** pie charts displaying hit rates in sets presenting three CMAs. In all trials, the reference sound was consistently a "coo," but the match in each session was one of the four monkey pictures. Hits are depicted in colors, while false alarms (FAs), occurred when the monkey chose a non-matching picture, are shown in gray or white. **Right panel,** reaction time (RT) distributions of hits are illustrated with the same color coding as in the left panel. **Inset,** FA distributions produced in trials where one of the four monkey pictures was presented as a match, but a picture of a 'human' or a 'cow' was selected. (C) Same format as **B** but for 'cow' CMAs. (D) The standard deviations (STDs) of the RT distributions increased as a function of their means during hits, false alarms (FAs), and in trials with two, three, or four pictures on the screen. (E) Plot of the monkeys' HRs as a function of the mean RTs of hit distributions in **D**.

pictures) used at this experiment rather than in the number of new sounds (**Fig 4C**). **Fig 4D** shows that Monkey M presented a similar effect in 3 CMAs, performing above 50% chance in trials of only two pictures on the touchscreen (i.e., 72.22% of the versions; paired-sample t-test, p < 0.05) and distributing RTs by picture category (**Fig 4E**), further supporting the notion of auditory invariance. In other words, regardless of variations in sounds, the animals could recognize them. Altogether, our findings suggest that monkeys can perform CMAs based on the ability to perceive equivalences within different sounds of the same type.

## Discussion

To investigate if rhesus monkeys can associate sounds with images regardless of their ethological relevance, we engaged two of these primates in a DCMMS task. To solve the task, monkeys had to retain in WM either an auditory replay or a crossmodal equivalent of the sounds (i.e., a face) and compare the memory against different pictures to find the match. Evaluation of their performance across various tests yielded two main outcomes: 1) the monkeys adeptly formed associations between sounds (e.g., animal vocalizations, words) and pictures (e.g., faces,

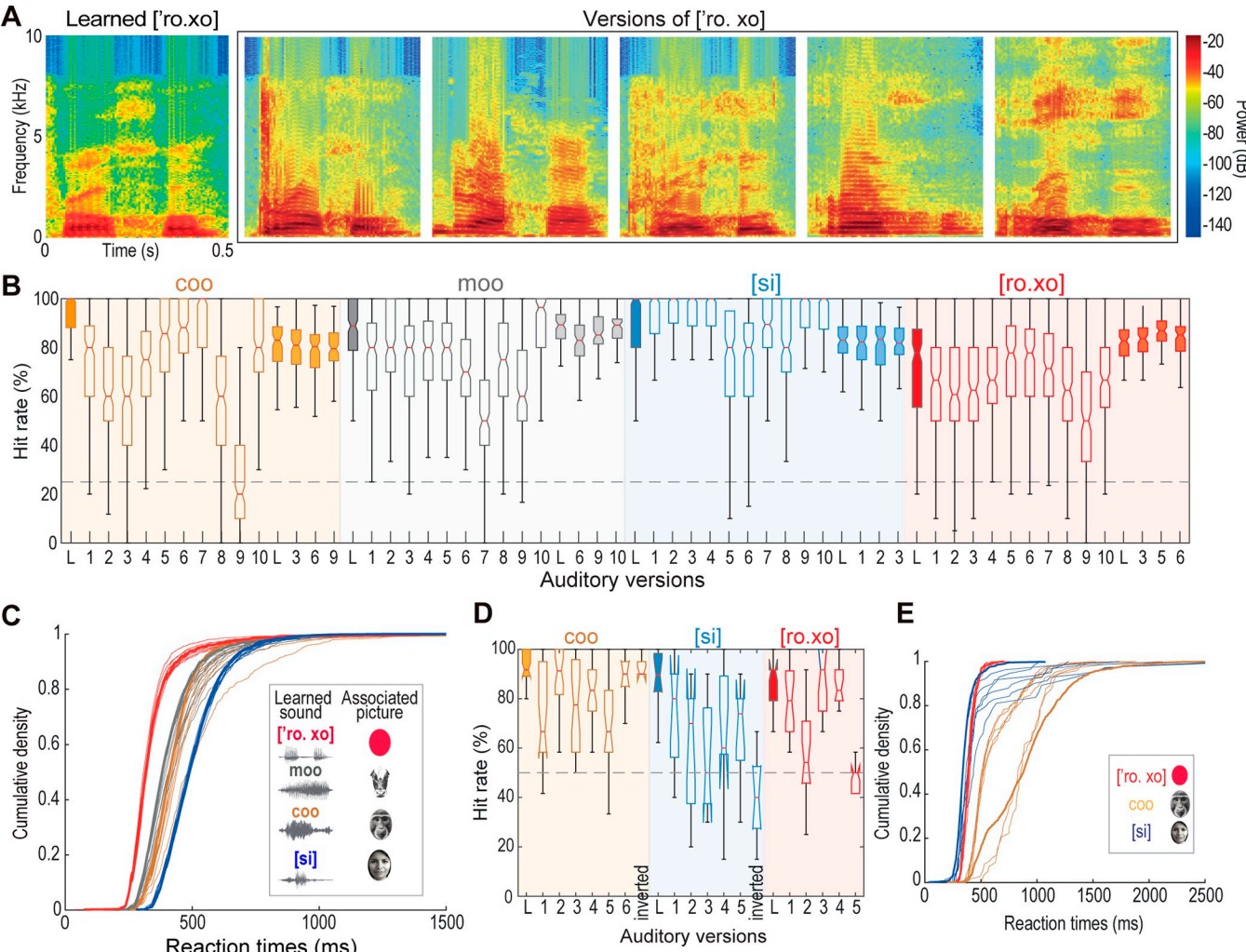

**Fig 4. Monkeys recognized sounds uttered by different individuals. (A)** Spectrograms from various speakers depicting the Spanish word ['ro. xo] (red). The spectrogram of the learned sound is on the left. **(B)** Hit rate of monkey G in all sounds' versions. Closed boxes on the left represent HR in the learned sounds (L). Open boxes, different versions' HR. Closed boxes on the right of each group correspond to the HR in versions comprised of double repetitions of some sounds including L. **(C)** Cumulative density functions of the RTs in the learned sounds (bold lines) of monkey G and their versions. Notice how the distributions group by the picture category rather than by sounds. **(D)** Same as in **B**, but for monkey M. **(E)** Same as **C**, but for monkey M. The pictures are similar but not identical to the original images used in the study and are therefore for illustrative purposes only.

cartoons), demonstrating human-like word-object associations that form the basis of language (**Figs 1 and 2, S1 Table**), and 2) these associations generalized even when the vocalizations and words they learned were uttered by different voices (**Fig 4, S4 Table**). Subsequent sections will detail these findings and explore the potential mechanisms to establish CMAs.

## Rhesus macaques create crossmodal associations between sounds and images of different types

Previous studies demonstrated that monkeys could perform crossmodal discriminations of supramodal information such as numerosity and flutter frequencies [50–54] and learn and group numerous sounds into categories irrelevant to their ethology [55, 56]. However, establishing cross-modal associations between NER categories in monkeys has proved to be challenging [37, 46–48]. In training two rhesus monkeys in the DCMMS task, we initially

encountered hurdles as the monkeys tended to disregard sounds [64, 65]. To counter this, training began with sound detection and progressively moved to crossmodal associations. We obtained different learning parameters from the monkeys' performances in each CMA across sessions (**Fig 2**).

During the initial training phase, the monkeys learned to interact with the task's apparatus (i.e., pressing the lever and touching the screen), achieving controlled motor responses within one or two weeks. Learning the first CMA (i.e., a broadband noise paired with a cow cartoon) required many sessions. Subsequent CMAs achieved statistical performance in just a few sessions; however, the animals excelled at the task after many practice sessions. We found no clear evidence that learning CMAs that included possible ethologically relevant stimuli like human and monkey faces, or coos [20–31] were facilitated more than other CMAs to which they had no previous exposure. In other words, the animals learned all CMAs at similar rates, providing behavioral data that could be highly informative regarding the brain responses underlying CMAs. Future neurophysiological evidence could build on these behavioral findings.

Three of our results aligned with the idea that CMAs could be created from templates [5–12]: 1) monkeys learn each new CMA faster; 2) mastering a CMA requires a prolonged period, akin to learning to speak in humans; 3) the animals' performance remained consistently high when the same vocalizations or words were presented with different voices, suggesting that the acoustic variations activated auditory templates, similar to how formants in words trigger acoustic recognition in monkeys [55]. Similarly, our results suggest that visual templates could create perceptual equivalence among different faces of the same type (**Fig 3B**). This is the strongest evidence to date that supports the possibility that monkeys can connect auditory and visual templates as humans do.

The formation of supramodal circuits linking vocalizations with other motor behaviors [12, 13] has suggested that the integration process in NHPs might similarly involve motor and spatial associations across sensory modalities [66, 67]. In our task, such associations were unnecessary since the animals had to match a sound with the corresponding picture, which was presented at different locations every trial. Moreover, studies exploring the convergence of crossmodal information in WM [22–27, 48] indicate that while motor or spatial associations may facilitate initial learning, more abstract associations such as numerosity or flutter [50, 54], extending beyond immediate and innate categories, can be developed through direct CMAs. Therefore, the monkeys performing our task could have created direct connections between auditory and visual templates.

## Working memory mechanisms for crossmodal matching

In contrast to other tasks [21, 37], our monkeys had to retain information about sounds over a 3-second delay and use it to compare with different pictures until they found a match, similar to previous work on the intra- and cross-modal discrimination of flutter [51–54]. Given that the animals performed above chance in all CMAs, and strategies such as selecting a particular picture or location cannot explain their performance (**S2 Fig**), we conclude that the most parsimonious explanation was the cross-modal matching of sounds and pictures. In other words, monkeys must have retained information about the sounds in WM to find the cross-modal match presented 3 seconds later. A candidate brain region for the type of WM involved in our task is the PFC [19], which participates in the retaining of parametric and nonparametric information of different sensory modalities compared intra- or cross-modally [2, 3, 20–31]. Notably, the PFC is also responsible for intramodal associations of stimuli separated in space and time [50]. Therefore, it is probably capable of translating information cross-modally; in

our task, this could involve possibly invoking visual representations after hearing sounds, thus retaining visual information in WM for later comparisons with the pictures, rather than keeping the reference sound in working memory until the pictures appear.

On the other hand, it is well documented that PFC activity in the context of CMAs is activated by ethologically relevant stimuli such as conspecific faces and voices in monkeys not engaged in their active recognitions [26, 42]. This suggests that ethologically relevant circuits could be established there since birth [26, 31]. Therefore, active cross-modal discrimination and the learning of CMAs between non-ethological stimuli may occur in other areas of the temporal lobe, known to represent and integrate auditory and visual objects [37–45], showing activations to superimposed audiovisual stimuli [37], perhaps to facilitate the recognition of individuals within their social group [26]. However, only future neurophysiological experiments in monkeys trained in the DCMMS task would reveal not only how and where in the brain non-ethological auditory and visual categories are learned, stored, and associated cross-modally, but also whether auditory or visual images invoked by sounds are retained in WM during the resolution of the task.

## Supporting information

**S1 Video. Monkey G performing the DCMMS task.**
(MP4)

**S1 Table. Monkeys' learning parameters and hit rate.**
(PDF)

**S2 Table. Overall hit rate (mean ± STD) in four CMAs.**
(PDF)

**S3 Table. The proportion (mean ± STD) of pictures selected.** Selections of pictures during hits and FAs in the condition when one sound was associated with different pictures of the same type.
(PDF)

**S4 Table. Hit rate (mean ± STD) in different versions of the learned sounds.**
(PDF)

**S1 Fig. Learning of CMAs.**
(PDF)

**S2 Fig. Hit rate and reaction times at different picture locations.** To analyze biases toward selecting a P at any angle from the center of the touchscreen, we performed a one-way ANOVA, False Discovery Rate corrected for multiple pairwise comparisons. Monkey M showed no location bias (p-values > 0.034). Monkey G, however, exhibited a significant effect for the monkey face position (F [15, 160.67] = 1.97; p = 0.014) and the cow face (F [15, 150.619] = 2.51; p = 0.001), but not for the human (p = 0.988). Post-hoc analysis (Tukey's HSD) revealed these differences occurred in angles < 90° within each screen quadrant. In other words, while there were biases in selecting pictures at angles, there was no consistent preference for a specific quadrant. Based on these findings, the behavioral results presented here correspond to subsequent experiments presenting pictures only in four quadrants.
(PDF)

## Acknowledgments

We extend our gratitude to Vani Rajendran for valuable feedback; Francisco Pérez, Gerardo Coello, and Ana María Escalante from the Computing Department of the IFC; Aurey Galván

and Manuel Ortínez of the IFC workshop; and Claudia Rivera for veterinary assistance. Additionally, we thank Centenario 107 for their hospitality.

## Author Contributions

**Conceptualization:** Luis Lemus.

**Data curation:** Elizabeth Cabrera-Ruiz, Marlen Alva, Miguel Mata-Herrera, Tonatiuh Figueroa.

**Formal analysis:** Elizabeth Cabrera-Ruiz, Marlen Alva, Mario Treviño, Miguel Mata-Herrera, José Vergara, Luis Lemus.

**Funding acquisition:** Luis Lemus.

**Investigation:** Elizabeth Cabrera-Ruiz, Miguel Mata-Herrera.

**Methodology:** Elizabeth Cabrera-Ruiz, Luis Lemus.

**Software:** Elizabeth Cabrera-Ruiz, Marlen Alva, Mario Treviño, Tonatiuh Figueroa, Luis Lemus.

**Validation:** Mario Treviño, José Vergara, Javier Perez-Orive, Luis Lemus.

**Visualization:** Elizabeth Cabrera-Ruiz, Marlen Alva, Luis Lemus.

**Writing – original draft:** Luis Lemus.

**Writing – review & editing:** Elizabeth Cabrera-Ruiz, Marlen Alva, Mario Treviño, José Vergara, Javier Perez-Orive.

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
