## [Decision Letter · Decision Letter 0]

19 Aug 2024

PONE-D-24-30219

Crossmodal association between auditory and visual objects in rhesus monkeys

PLOS ONE

Dear Dr. Lemus,

Thank you for submitting your manuscript to PLOS ONE. After careful consideration, we feel that it has merit but does not fully meet PLOS ONE’s publication criteria as it currently stands. Therefore, we invite you to submit a revised version of the manuscript that addresses the points raised during the review process.

Please read carefully the comments posed by the reviewers and that will also address any specific concerns from my side.

We look forward to receiving your revised manuscript.

Kind regards,

Argiro Vatakis

Academic Editor

PLOS ONE

Journal Requirements:

4. We note that Figures 1-4 ans S3 includes an image of a participant in the study. 

Reviewers' comments:

Reviewer's Responses to Questions

**Comments to the Author**

1. Is the manuscript technically sound, and do the data support the conclusions?

Reviewer #1: Partly

Reviewer #2: Yes

2. Has the statistical analysis been performed appropriately and rigorously? 

Reviewer #1: Yes

Reviewer #2: Yes

3. Have the authors made all data underlying the findings in their manuscript fully available?

Reviewer #1: Yes

Reviewer #2: Yes

4. Is the manuscript presented in an intelligible fashion and written in standard English?

Reviewer #1: Yes

Reviewer #2: Yes

5. Review Comments to the Author

Reviewer #1: The manuscript of Cabrera-Ruiz is focused on macaques‘ abilities to learn audio-visual crossmodal associations in a delayed match-to-sample task.

Although the text is generally very well written, in my opinion the main focus/emphasis of the paper is not stated clearly enough. As far as I understand, one of the main achievements of the study is the new procedure for training macaques to perform the audio-visual cross modal task. However instead of emphasizing that achievement and discussing it properly, the authors extensively comment on (arbitrary) proposed models of cross-modal associations, which cannot be tested without application of electrophysiological methods.

Major comments:

Introduction: More information (including existing models) about cross-modal association in macaques should be provided.

Line 42: The authors mention that the subjects are relatively old, which makes me think that they have experience in many other tasks. It would be good to comment whether and in what way the previous tasks, which the subjects were trained in, could affect their ability to learn the crossmodal associations.

Line 49: What would be the influence of the environmental enrichment to the performance of the audio-visual task?

Line 94-102 – Monkey Training: In my opinion this is one of the most important methodological part of the paper. More details should be provided regarding the training procedure.

Line 146 / Fig 2C, insets: It would be interesting not just to present the FA rates, but also to show which are the most common mistakes (e.g. are monkey faces confused more often with human faces in comparison to the red oval).

Lines 188-195: The authors identify 3 key findings, the first 2 of which directly follow the presented results. The third finding claims that “the monkeys’ success in forming crossmodal associations required […] working memory”, which is logical, however it is unclear how it could be derived from the results.

Lines 208-212: In my opinion this is the most important contribution of the study and I think that it should be discussed more extensively.

Lines 232-270: The two presented models are very interesting, however in the context of this behavioral experiment they are completely arbitrary. I am not sure whether they should be discussed so extensively without any electrophysiological data which can support either of them.

Minor comments:

Line 2: the sentence starting with “To find out” is ambiguous. To find out what?

Line 98: probably “cue to” instead of “clue for”.

Reviewer #2: Review for “Crossmodal association between auditory and visual objects in rhesus monkeys”

PONE-D-24-30219

This paper assessed whether rhesus macaques can learn crossmodal associations, how they are perceiving the visual sets versus the auditory sets in those associations, and what models of visual or auditory memory might explain these patterns. They successfully trained two macaques on a set of crossmodal associations (CMAs), which is no trivial feat. Performance on the task is well presented and clearly above chance. Upon further assessment, macaques appear to treat different individual sounds of the same type as perceptually equivalent, whereas patterns of reaction times for different images suggests this might not be the case for the visual image sets. This has interesting implications for how macaques may form these types of representations, and the authors put forth some possible models for how these representations might be stored in memory to explain these patterns.

I believe this experiment is interesting and relevant to ongoing work in the field. The data is compelling, and the experimental design effectively assesses their questions. There is room for improvement in how the paper is framed, as well as in the clarity of some of the explanations and data presentation, but I believe the work itself is sound, and will be useful groundwork for future studies in the field. I recommend this paper for publication, pending some revisions based on my suggestions below.

General Suggestions

1) Framing of memory systems in the intro. Your intro only talks about assessing whether macaques can learn CMAs. This makes sense since that’s a major point of your study, but this paper isn’t really about the CMA learning itself. I spent most of the paper thinking about the learning and acquisition of CMAs, since that’s how it’s framed here, but you don’t really make any points about acquisition at all. So much of your discussion is dedicated to how those CMAs are represented and maintained in the brain, and using macaques as a model system for assessing these different possible neural models. You need to lay the groundwork for that here. Are there existing competing hypotheses about how CMAs are represented in humans or macaques that you could discuss here? At the very least you need to mention that CMAs can be held in working memory, or that memory is relevant to this process at all. Even some basic background about audio versus visual working memory and how they might be relevant here would help readers keep that framing in mind and not be blindsided by the discussion.

2) How you’re referring to CMAs and sets of CMAs. It took me a while to figure out how exactly your different, specific CMAs were grouped (i.e. one single coo was trained to be associated with a few different monkey images, and each of those individual associations was one CMA). It would be useful to clarify this early, and possibly give name to the different CMA type groups (i.e. CMA sets or something like that), because knowing the associations were one coo to multiple images changes how a reader might interpret the results. I know you added more coos later on, but I spent the majority of the paper thinking each CMA was made up of completely different individual sounds and images. Even the S3 table didn’t clarify this for me, as I read the list as being grouped into categories (i.e. “coos”) with the individual pictures presented, and I interpreted the letters next to each picture as specifying the name of the specific sound used with that picture, rather than as the identifying name of the picture itself.

3) CMA acquisition. Do you have acquisition data for every time a new CMA is added? It sounds like once monkey M initially learned a CMA type (the coo, for example), they didn’t take much training to learn to associate a new image with that coo (I believe you added in new individual CMA images gradually, correct? If that’s not the case then I’m misinterpreting the methods and they should be clarified). If they are in fact generalizing that single “coo” learning to new images, that is pretty compelling evidence for a learned label (“coo” means “choose monkey”, for example) rather than many separate individually learned associations. That seems like really relevant information for how these CMAs are acquired and represented, and would definitely be worth adding. You could plot the number of presentations they took to learn each new CMA image, and if they’ve really learned a category label, I would expect those numbers to drop drastically for CMAs acquired later.

4) Other crossmodal association literature. There is a fair amount of work looking at crossmodal associations in humans and animals, though it tends to be framed more towards identity recognition, that would be worth looking into and including. Here’s a useful review:

Perrodin, C., Kayser, C., Abel, T. J., Logothetis, N. K. & Petkov, C. I. Who is That? Brain Networks and Mechanisms for Identifying Individuals. Trends in Cognitive Sciences 19, 783–796 (2015).

5) Arbitrary learned associations, or existing associations between animals and their calls? Because I was thinking so much about learning, I had a specific running question throughout the paper, which is whether the macaques are actually learning arbitrary associations, or coming in with existing, ecologically relevant associations of certain sounds with certain animals. I think this would be easy to fix with some additional support. If you think they’re arbitrary (and I think I agree with you), you just need to provide evidence. For example, if you have reason to believe the macaques don’t perceive faces on screen as though they’re actual macaques/humans/etc, or that you saw similar patterns even with your cartoon stimuli and the color red, or that they’ve never seen or heard cows before, or something along those lines.

Specific Suggestions

Abstract:

Line 1: I would be careful here. Your topic of interest is symbolic crossmodal associations, but even with this cool task, you can’t really make claims that your macaques have learned something symbolic, especially since it’s a unidirectional association and you never test the reverse (i.e. see image choose sound, which would be a bigger methods challenge). Your first line is fine, because it’s true that it is uncertain, but in the second line you need to make it clear that you’re testing for a step below symbolism, just any crossmodal associations at all, and not making any claims about whether they’re represented symbolically in macaques.

(see these references for more info on symbolic representations:

Palmer, F. R., & Palmer, F. R. (1981). Semantics. Cambridge University Press.

Deacon, T. W. (1997). The symbolic species: The co-evolution of language and the brain (1st ed). W.W. Norton.)

Line 6: You say performance remained constant, but I think that undercuts you a bit. They did well, it wasn’t just constant, it was consistently high.

Line 9: You say “semantic and conceptual thinking at the single-neuron level,” which doesn’t really say specifically what you mean and also is never mentioned anywhere else in the paper. I would shift this to refer to what you actually conclude they could be a model for, which is, from what I can tell, the neural pathways by which semantic information stored in memory, and later applied.

Intro:

Line 18: Can you add a reference from human literature to support this claim? It’s not controversial, but there is SO much research on how words are acquired in children, it seems odd to leave it out.

Line 28: See first general suggestion about framing.

Materials and Methods:

Line 77: Stims were presented in an equidistant circle, but did the circle rotate? Was configuration within the circle randomized? How was the actual position and configuration decided?

Line 84: Specify that this is just a cue to denote the start of the trial, I was confused for a moment thinking you were presenting the actual picture before the sound was played.

Line 88: This would be a good place to clarify the specifics of the CMA sets.

Line 100: Here for example, when you say they learned new CMAs, were they all the same type? As in they started out differentiating coos and human words, and the additional CMAs were new images corresponding to coos and words? Or were they totally new CMA types, as in the addition of moos?

Results:

Line 119: In S2 Fig B, there are 3 clear dips in performance that I imagine correspond to the addition of a new CMA type/additional foil stimulus, is that right? You should mark where those additions occur on the graph. Unless I’m misinterpreting and this is plotting performance once all 4 CMA types have been added, in which case that should be clarified.

Line 156: They seem to have learned a category set. Coo means touch monkey. So the thing that would slow them down is potentially just how long it takes to recognize that the individual monkeys are monkeys, leading to the reaction time differences (Fig. 2 A and C). When you gave them new images with the sounds, did they generalize to new monkey pictures? This is where the acquisition data would be really useful.

In Fig 2D, it seems the takeaway is that when average RTs are longer their variance is higher, is that correct? So there’s more variance in RTs when they’re taking longer to decide. I wasn’t sure what this graph added until I got to the conclusions, because it seems this pattern is the evidence you’re citing for the visual imagery model. I don’t fully see why this is evidence for that model, so I could use more clarity on that front overall. As it stands I’m not sure that I see the importance of this figure.

Line 172: I would clarify that perceptual invariance is not actually an inability to perceive a difference between the sounds. The critical thing is that they are treating these sounds as equivalent (the way we treat a word as the same no matter whose voice is saying it, or a musical note as the same even if it’s in a different octave), but that they would still likely be able to differentiate the individual sounds if need be. That is an important distinction.

Possible follow-up thought, I wonder where macaques draw these category lines. Would they view humans saying different words as equivalent, or would different words each have their own set? What about monkey coos vs screams, presumably those would fall into different categories, but could they learn to lump them together into categories of “monkey” and “human” instead?

Discussion:

Line 191: A point where you could defend that you think these associations are arbitrary.

Line 193: This memory conclusion is surprising, would be less jarring if more theoretical background on this is added to the intro.

Line 220: Do you report this data anywhere?

Line 224-230: This explanation helped me understand why you included figure 2, I would try to bring some of this clarity up above.

Line 232: Interesting hypotheses being put forth but again comes out of nowhere, add better framing for it in the intro.

Line 261: How so? What makes the patten you see a better match for the visual model? Don’t both models require some amount of processing and decision making once the visual stimuli are presented, resulting in differences in RT based on the specifics of the image? Arguably converting an image to a sound to match against your representation of a sound would in fact require more processing at image presentation than matching an image directly to your existing image representation, so wouldn’t the auditory model potentially result in longer and more varied processing once the choice images are presented? Maybe I’m missing something, but it would be useful for you to explain to the readers why you think this model is a better match to your data.

6. PLOS authors have the option to publish the peer review history of their article (what does this mean?). If published, this will include your full peer review and any attached files.

Reviewer #1: **Yes: **Ivo D. Popivanov

Reviewer #2: No

---

## [Author Response · Author response to Decision Letter 0]

9 Nov 2024

Dr. Argiro Vatakis

Dear Prof. Vatakis,

We are pleased to submit a revised version of our manuscript, "Crossmodal Association between Auditory and Visual Objects in Rhesus Monkeys" (PONE-D-24-30219), which addresses all the concerns and suggestions raised by the reviewers, as well as editorial requests. We have modified the manuscript, figures, and figure captions to meet the publication criteria of PLOS ONE.

We believe the updated manuscript, enhanced by the reviewers' input, effectively demonstrates its contribution to the understanding of learning mechanisms in cross-modal associations and their implications in sensory processing and working memory. The revisions feature an introduction reframed to emphasize working memory and cross-modal associations in macaques, together with comprehensive methodological details on training the macaques and analyses of their performance during various learning stages. Additionally, we updated the title to "Monkeys Can Identify Pictures from Words," capturing a crucial insight from the reviewers regarding the learning of cross-modal associations between non-ethological stimuli. To address concerns about the monkeys' learning processes, we included the former S2 Fig as a new Fig 2. We have also incorporated supplementary information and new panels within the original figures to better address the reviewers’ comments and clarify the findings. We also withdrew the previous Fig 4 from the manuscript since it depicted possible models for solving cross-modal associations in the brain, which were not tested in our experiments. Nevertheless, all the results presented in the initial draft are maintained, enriched only by deeper interpretations suggested by the reviewers. We confirm the completeness and correctness of the reference list, with a few references added to the introduction to align with suggestions on the study framing.

We also confirm that all the figures in the manuscript (Figs 1-4, S1, S2 and S3) were created entirely in our lab. Some of them appear as a preprint in bioRxiv.org (https://doi.org/10.1101/2024.01.05.574397), but none of them have been previously copyrighted. The original images used in the experiments were downloaded from free online sites and the original version of the manuscript depicted them. However, we replaced all those depictions from all figures to avoid any possible copyright conflicts, and substituted them with pictures similar but not identical, created using an AI image generator (https://www.fotor.com/ai-art-generator). Fotor.com explicitly states that: "For the AI image generator, the AI-generated images can be used for both personal and commercial purposes. You can share them on social media platforms, use them for marketing campaigns, or sell them. You are the copyright owner of your creations and will be responsible for any output that you generate using AIGC Related Service." (https://support.fotor.com/hc/en-us/articles/17767970123417-Are-the-AI-generated-images-commercially-available-Do-I-have-ownership-of-them). Moreover, as requested by PLOS ONE, we included the following sentence in figure captions: “The pictures are similar but not identical to the original images used in the study and are therefore for illustrative purposes only.” We also included the following description at the Methods section (lines 109-113): “Animal pictures used in the experiment were downloaded from free online sites and customized. However, the pictures shown in figures and supplementary information are similar but not identical to the original images used in the study; they were created for illustrative purposes only using an online AI image generator (https://www.fotor.com/ai-art-generator).” However, we included the picture of one of the researchers in the study. Therefore, we submitted the signed consent format as provided by PLOS ONE, and report at the Ethics statement (lines 65-66) the following: “The portrayal of one of the authors of this manuscript was used in the experiments and has given written informed consent to publish this case details.” 

Regarding the sounds used in our experiments, we confirm that the cow vocalizations were downloaded from Freesound.org, and under the following License: Creative Commons 0 (CC0; No copyright). However, all the human and monkey sounds were recorded in our lab and have not been published elsewhere. We clarify this at lines 99-101 of the methods as follows: “The experiment utilized a variety of sounds, including laboratory recordings of words and monkey vocalizations, as well as free online sounds of cow vocalizations (https://freesound.org/).”

Finally, we were notified that the Funding Information, Financial Disclosure and Data Availability statements need to be clarified. Therefore, we kindly ask to include the following statements: 

Funding Information

LL received grant support from the Consejo Nacional de Humanidades Ciencias y Tecnologías (CONAHCYT; Number: 256767; https://conahcyt.mx/), and the Programa de Apoyo a Proyectos de Investigación e Innovación Tecnológica (PAPIIT; Number: IN229323; https://dgapa.unam.mx/index.php/impulso-a-la-investigacion/papiit). JV received support by Secretaría de Educación, Ciencia, Tecnología e Innovación de la Ciudad de México (SECTEI/103/2022; https://www.sectei.cdmx.gob.mx/). Elizabeth Cabrera Ruiz conducted this study to fulfill the requirements of Programa de Doctorado en Ciencias Biomédicas of Universidad Nacional Autónoma de México and received a doctoral scholarship from Consejo Nacional de Humanidades Ciencias y Tecnologías (Scholarship number: 245771; https://conahcyt.mx/). The data in this work are part of her doctoral dissertation.

Financial Disclosure

Data Availability Statement

The data is fully available without restrictions from Figshare.com https://figshare.com/s/998043a02c8b15315632

Competing Interests

No authors have competing interests

Author contributions

Conceptualization: LL. Data Curation: ECR, MMH, TF. Formal Analysis: ECR, MA, MT, MMH, JV & LL. Funding Acquisition: LL, ECR, JV. Investigation: ECR, MMH. Methodology: ECR & LL. Software: ECR, MA, TF, MT, & LL. Validation: JPO, MT, JV & LL. Visualization: ECR, MA & LL. Writing original draft: LL. Review & draft editing: ECR, MA, JPO, MT & JV. 

Responses to the reviewers (highlighted in blue)

Reviewer #1: 

The manuscript of Cabrera-Ruiz is focused on macaques‘ abilities to learn audio-visual crossmodal associations in a delayed match-to-sample task.

Although the text is generally very well written, in my opinion the main focus/emphasis of the paper is not stated clearly enough. As far as I understand, one of the main achievements of the study is the new procedure for training macaques to perform the audio-visual cross modal task. However instead of emphasizing that achievement and discussing it properly, the authors extensively comment on (arbitrary) proposed models of cross-modal associations, which cannot be tested without application of electrophysiological methods.

The revised manuscript now emphasizes the achievement of training macaques in cross-modal discrimination. For example, line 8 of the Abstract now reads: "We found that the monkeys learned and performed proficiently in over a dozen associations." We also included full descriptions of the training protocol as a "Monkeys training" subsection at the Methods (lines 146-177), and a "Learning measurements" subsection about the analytical methods for evaluating the learning process (lines 178-203). In addition, we included a new section "Rhesus Monkeys Can Learn Cross-Modal Associations Between Stimuli of Different Types" at the Results (lines 251-294), and included S2 Fig in the text as Fig 2 (lines 277-282) with better descriptions of the learning of CMAs across training sessions, and a new panel at Fig 2B (also suggested by Reviewer 2), showing the number of sessions to performing each CMA above chance. Finally, we created a new S1 Fig showing the performance for each CMA across sessions.

We also agree with the reviewers about the problem of presenting models about the neuronal mechanisms of CMAs without evidence. Therefore, we decided to remove Figure Four from the document, limiting ourselves exclusively to adding information about cross-modal association models in the introduction. We now refer to the acoustic template model as a mechanism for association between modalities and to the alternative model of learning through social interactions. These models are important for the association between modalities as a basis for associations between stimuli that are not necessarily ethological but that are used in humans for language (which was what we tried to test in our study). The first paragraph of the introduction now reads as follows:

“Humans form cross-modal associations (CMAs) between sounds and images, which play a vital role in integrating semantic representations within language [1]. Supporting this, fMRI studies have shown that the temporal lobe of the human brain is actively involved in CMAs [2,3] between words and visual objects [4]. It is believed that CMAs between phonological "templates"—developed in human infants by listening to caretakers—and observed objects are essential for creating semantic representations and aiding the production of a child's first words [5–8]. Similarly, auditory templates have been proposed as a mechanism for vocal production in birds [9–13] and marmoset monkeys [14]. Recent studies, such as those by Carouso-Peck and Goldstein [15,16], have also shown that visual signals during social interactions can also influence vocal production in birds. However, only a few ethological studies have suggested the existence of CMAs between vocal sounds and visual cues for semantic communication [17]. For instance, research has observed that vervet monkeys respond to calls signaling the presence of predators by looking upwards, downwards, or climbing into trees [18].”

Major comments:

Introduction: More information (including existing models) about cross-modal association in macaques should be provided.

We included important references to cross-modal association processes in monkeys. Importantly, we also framed the introduction within the scope of working memory since it plays a key role in multisensory association processes (as noted by reviewer 2). The second paragraph of the introduction now reads as follows:

“Neurophysiological recordings in monkeys have shown that the prefrontal cortex (PFC) — a brain area homologous to that in humans — utilizes working memory (WM) circuits [19] to perform CMAs between voices and faces [20–32], receiving inputs from various sensory regions [33–36]. CMAs have also been observed in the auditory and visual areas of the temporal lobe [37–45]. Notably, trained macaques have demonstrated the ability to perform cross-modal discriminations between visual and tactile objects [46,47], and between stimuli that could be considered non-ethologically relevant (NER), such as between pitch and color [48] and between amodal information (i.e., information that does not belong to a particular modality) [49] such as numerosity [50] and flutter frequencies [51–54]. However, it remains to be explored whether non-human primates can establish CMAs between NER stimuli that are important for human language, like words —which monkeys can discriminate phonetically [55,56], and pictures.”

 Line 42: The authors mention that the subjects are relatively old, which makes me think that they have experience in many other tasks. It would be good to comment whether and in what way the previous tasks, which the subjects were trained in, could affect their ability to learn the crossmodal associations. 

The monkeys had not participated in any experiments previously nor had they been trained in any task. We now clarify this in the Methods on line 69, which now reads as follows:

“The animals had no previous training in any other task and were not subjected to any surgery or head restraint for this behavioral study.”

 Line 49: What would be the influence of the environmental enrichment to the performance of the audio-visual task? 

 Environmental enrichment was implemented as part of the animals' housing, e.g., plastic boxes with snacks inside to promote manual dexterity. Additionally, the monkeys spent time in a shared space with room to climb and socialize with other monkeys, which apart from favoring the display of social behaviors characteristic of their species, probably played a role in the 'monkey' CMA. Similarly, the animals were partially exposed to the faces of the researchers (i.e., they wore masks), and only the picture of the authors was used as a stimulus. Therefore, we believe that it is unlikely it helped the monkeys significantly in creating the associations for the task. The lines 77-82 address enrichment as follows:

“The monkeys also had access to an enriched environment with toys, a recreation area for climbing and socializing with other monkeys four days a week, and opportunities for grooming through mesh sliding doors. In addition, cartoons and wildlife videos of content unrelated to the experiments, were presented on TV for no more than four hours a day. However, the face and voice of one of the researchers with whom the monkeys interacted were used during the experiments”. In this regard, line 393 of the discussion reads: “We found no clear evidence that learning CMAs that included possible ethologically relevant stimuli like human and monkey faces, or coos [20–31] were facilitated more than other CMAs to which they had no previous exposure.” 

 Line 94-102 – Monkey Training: In my opinion this is one of the most important methodological part of the paper. More details should be provided regarding the training procedure.

 We expanded the Methods section ("Monkeys Training," lines 146-177) to include a detailed description of the training protocol in the Methods section ("Learning Measurements," lines 178-203), where we describe analytical methods of the monkeys’ learning process. We incorporated S2 Fig as Fig 2 so the monkeys' learning of CMAs can now be observed—a critical finding in our study. We also included S1 Fig to show the performance at all CMAs throughout learning and added sections to the Results and the Discussion specifically addressing the learning of CMAs. Specifically, sections "Rhesus Monkeys Can Learn Cross-modal Associations Between Stimuli of Different Types" (lines 251-294), and "Rhesus Macaques Create Crossmodal Associations Between Sounds and Images of Different Types" (lines 377-417), respectively. 

Line 146 / Fig 2C, insets: It would be interesting not just to present the FA rates, but also to show which are the most common mistakes (e.g. are monkey faces confused more often with human faces in comparison to the red oval).

Addressing this question, we included pie charts in Figure 3 (formerly Figure 2), showing the pictures selected during false alarms. We also included better descriptions of the results (lines 316-331) and a new S3 Table with the percentage of choices during hits and false alarms.

Lines 188-195: The authors identify 3 key findings, the first 2 of which directly follow the presented results. The third finding claims that “the monkeys’ success in forming crossmodal associations required […] working memory”, which is logical, however it is unclear how it could be derived from the results.

Regarding working memory, our task required it due to the 3s delay between auditory stimuli and picture options. We now provide better descriptions in the abstract, introduction, methods, results, and discussion. The abstract (lines 6-8) now reads: “In each trial, the monkeys listened to a brief sound (e.g., a monkey vocalization or a human word), and retained information about the sound to match it with one of 2–4 pictures presented on a touchscreen after a 3-second delay.” The Introduction (lines 45-50) now reads: “We specifically designed the task to temporally separate auditory and visual stimuli, thus engaging WM circuits to retain one modality in mind while awaiting the corresponding cross-modal stimulus.

---

## [Decision Letter · Decision Letter 1]

23 Dec 2024

Monkeys can identify pictures from words

PONE-D-24-30219R1

Dear Dr. Lemus,

We’re pleased to inform you that your manuscript has been judged scientifically suitable for publication and will be formally accepted for publication once it meets all outstanding technical requirements.

Kind regards,

Argiro Vatakis

Academic Editor

PLOS ONE

Additional Editor Comments (optional):

Reviewers' comments:

Reviewer's Responses to Questions

**Comments to the Author**

1. If the authors have adequately addressed your comments raised in a previous round of review and you feel that this manuscript is now acceptable for publication, you may indicate that here to bypass the “Comments to the Author” section, enter your conflict of interest statement in the “Confidential to Editor” section, and submit your "Accept" recommendation.

Reviewer #1: All comments have been addressed

Reviewer #2: All comments have been addressed

2. Is the manuscript technically sound, and do the data support the conclusions?

Reviewer #1: Yes

Reviewer #2: Yes

3. Has the statistical analysis been performed appropriately and rigorously? 

Reviewer #1: Yes

Reviewer #2: Yes

4. Have the authors made all data underlying the findings in their manuscript fully available?

Reviewer #1: Yes

Reviewer #2: Yes

5. Is the manuscript presented in an intelligible fashion and written in standard English?

Reviewer #1: Yes

Reviewer #2: Yes

6. Review Comments to the Author

Reviewer #1: In my view the manuscript is substantially improved with respect to the previous version. I don't have any additional comments and I would be glad to see that this well performed study will be published in this journal! Good luck with your further research on this topic!

Reviewer #2: All comments have been addressed, I believe this manuscript should be accepted for publication to PLOS One.

7. PLOS authors have the option to publish the peer review history of their article (what does this mean?). If published, this will include your full peer review and any attached files.

Reviewer #1: **Yes: **Ivo D. Popivanov

Reviewer #2: No

---

## [Editor Report · Acceptance letter]

10 Jan 2025

PONE-D-24-30219R1 

PLOS ONE

Dear Dr. Lemus, 

I'm pleased to inform you that your manuscript has been deemed suitable for publication in PLOS ONE. Congratulations! Your manuscript is now being handed over to our production team.

Kind regards, 

on behalf of

Dr. Argiro Vatakis 

Academic Editor

PLOS ONE